# Role of Phosphoinositide 3-Kinase in Regulation of NOX-Derived Reactive Oxygen Species in Cancer

**DOI:** 10.3390/antiox12010067

**Published:** 2022-12-28

**Authors:** Ali A. Akhiani, Anna Martner

**Affiliations:** Department of Infectious Diseases, Institute of Biomedicine, Sahlgrenska Academy, University of Gothenburg, 41346 Gothenburg, Sweden

**Keywords:** NOX, reactive oxygen species, PI3K pathway, cancer

## Abstract

Activation of NADPH oxidases (NOX) and the ensuing formation of reactive oxygen species (ROS) is a vital aspect of antimicrobial defense but may also promote tumorigenesis. Enhanced NOX activity has been associated with aberrant activation of oncogenic cascades such as the phosphoinositide 3-kinase (PI3K) signaling pathway, which is upregulated in several malignancies. In this review, we examine the role of PI3K on the regulation of NOX-induced ROS formation in cancer.

## 1. Introduction

Signaling via the phosphoinositide 3-kinase (PI3K) pathway controls a multitude of cellular processes including cell growth and survival, metabolism, migration, differentiation and proliferation. In addition, PI3K signaling may promote angiogenesis and the recruitment and activation of inflammatory cells [1]. Activation of PI3K is tightly related to tumorigenesis and every major node of this pathway is mutated in a variety of cancer forms. Among the most commonly mutated genes in cancer is *PIK3CA,* which encodes the p110 alpha catalytic subunit of class 1A PI3K, and the negative pathway regulator phosphatase and tensin homolog (*PTEN*) [2]. The most studied signal transduction protein activated downstream of PI3K, AKT (aka protein kinase B), regulates diverse cellular events including proliferation and survival [3,4].

Events triggered via the PI3K pathway entail activation of nicotinamide adenine dinucleotide phosphate (NADPH) oxidase (NOX) and production of NOX-derived reactive oxygen species (ROS) [5,6,7]. ROS are central to cellular redox regulation and exert positive feedback on PI3K singling by mechanisms including reversible oxidation and inactivation of PTEN along with inactivation of additional phosphatases that negatively regulate this pathway [7].

In addition to regulating redox-sensitive pathways, excessive amounts of ROS may cause cellular oxidative stress, genomic instability and, if released extracellularly, damage to surrounding cells [8,9,10,11,12,13]. An excess of cellular ROS, due to enhanced production and/or a deficiency in antioxidant defense systems, has been observed in many forms of cancer, including solid tumors such as prostate adenocarcinoma and melanoma along with hematopoietic malignancies [14]. Although previous studies have addressed the role of PI3K in cancer [1], the role of PI3K in regulation of NOX enzymes is only partially explored [15]. This review highlights recent advances in the regulation of ROS production via the PI3K pathway.

## 2. Class I PI3K Enzymes and Their Activation

PI3Ks comprise three main classes: class I, II and III; this classification is founded on their structural characteristics and substrate specificity [4,6]. The class I PI3K is most frequently implicated in human cancer [4] and the focus of this review. Class I PI3Ks are heterodimeric enzymes comprising a catalytic and a regulatory subunit and are further divided into subgroups IA and IB. The catalytic subunit of class IA comprises p110α, β or δ, encoded by *PIK3CA, PIK3CB* and *PIK3CD*, respectively, which dimerize with a shorter regulatory subunit (often p85) [16]. Class IB contains only the catalytic subunits p110γ, encoded by *PIK3CG*, and a regulatory unit p101 encoded by *PIK3R5*. Class I PI3K phosphorylates phosphatidylinositol-4,5-bisphosphate (PIP2) to generate phosphatidylinositol-3,4,5-trisphosphate (PIP3) [17]. This is likely the most important regulatory feature of class I PI3K, although these enzymes in vitro were shown to also convert phosphatidylinositol into phosphatidylinositol-3-phosphate, and phosphatidylinositol 4-phosphate into phosphatidylinositol-3,4-bisphosphate [18,19].

PI3K is activated by extracellular stimuli including growth factors, cytokines and hormones [20]. Activation of class I PI3Ks is induced by stimulation of receptor tyrosine kinases (for p110α, p110β and p110δ) or G protein-coupled receptors (GPCRs) (for p110β and p110γ) [21,22]. Activation of tyrosine kinase receptors involves the recruitment of the p85–p110 dimer to the membrane, which in turn releases the inhibitory effect of p85 on p110 to activate PI3K. The mode of activation by GPCR comprises direct interaction between the G protein Gβγ heterodimer and the PI3K subunits p110β, p110γ or p101 [23,24]. An activated p110 catalytic subunit phosphorylates PIP2 to generate PIP3. The PIP3 phosphoinositide domain is bound by proteins that contain Phox homology domains, pleckstrin homology domains (PH) or fab1/yotb/vac1/eea1 (FYVE) domains [25]. For example, AKT contains a PH domain that binds to PIP3, resulting in AKT translocation to the plasma membrane and activation. AKT in turn phosphorylates proteins including mTOR, which plays a central role in cancer initiation and progression [4]. PTEN regulates the levels of PIP3 by dephosphorylating PIP3 to PIP2 and thus negatively regulates the PI3K pathway [20].

## 3. Role of PI3K in Regulating the Tumor Microenvironment and Inflammation

Class I PI3Ks are involved in host defense by regulating phagocytosis, chemotaxis, ROS formation and cytokine production [26]. An important aspect of PI3K signaling in cancer is its role in promoting an inflammatory tumor microenvironment. While p110α and p110β are widely distributed in mammalian tissues, p110δ and p110γ are predominantly expressed in leukocytes [27]. PI3Kγ and PI3Kδ are thus the main isoforms implicated in regulating inflammatory responses to danger signals and antigens. Hence, PI3Kγ has been shown to be crucial for chemokine-mediated recruitment and activation of innate immune cells [26], while PI3Kδ is important for B and T cell receptor signaling, which is essential for lymphocyte development and survival [28]. A hallmark of B cell malignancies, including lymphoma and chronic lymphocytic leukemia (CLL), is the constitutive activation of B cell receptors (BCR). Idelalisib, a selective inhibitor of PI3Kδ, inhibits BCR signaling pathways critical for CLL cell migration and survival and thus promotes apoptosis in CLL cells [29,30]. Idelalisib in combination with therapeutic anti-CD20 antibodies such as rituximab improves progression-free survival in CLL and other B-cell malignancies but is associated with an increased risk of opportunistic infections [31], thus highlighting the role of PI3K in host defense [26].

In addition, angiogenesis, which is critical for tumor growth, is directly and indirectly regulated by PI3K signaling. Thus, PI3K activation in tumor cells and other cells in the tumor microenvironment may enhance secretion of proangiogenic factors such as VEGF and angiopoietins [32]. Stimulation of VEGF receptors and Tie receptors on endothelial cells by these inducers of angiogenesis activates PI3K/Akt-dependent endothelial cell proliferation, migration and differentiation [33].

Chronic inflammation is considered a risk factor for the evolvement of cancers and has also been shown to contribute to tumor growth and metastasis [34,35]. Inhibition of, in particular, PI3Kδ and γ signaling has been shown to reduce inflammation and severity of symptoms in models of autoimmune disease and allergic inflammation [36]. Inhibitors of class I PI3K may thus represent potential treatment options for non-resolving inflammatory diseases. Targeting the PI3K/Akt pathway is currently evaluated in various forms of cancer, as discussed in a recent review [4]. These inhibitors likely affect PI3K signaling in cancer cells as well as in cells within the tumor microenvironment.

## 4. NADPH Oxidases (NOX) and Their Activation

NOX enzymes are located in the plasma membrane or in membranes of intracellular organelles. The NOX enzymes catalyze the reduction of molecular oxygen to generate superoxide anion using NADPH as an electron donor [8]. Superoxide is spontaneously or enzymatically converted to hydrogen peroxide H_2_O_2_, which is the precursor for various bactericidal ROS including hydroxyl radicals and hypochlorous acid (HOCl) [37].

The first discovered enzyme of the NOX family, NOX2, is densely expressed on plasma membranes and on phagosomal membrane of myeloid cells, where NOX2 is pivotal in microbial killing [38]. Patients who are genetically NOX2-deficient thus encounter severe bacterial and fulminithic infections [39]. Six additional members of the NOX family of NADPH oxidases have subsequently been identified: NOX1, NOX3, NOX4 and NOX5 and dual oxidase (DUOX) 1 and 2, each with different subunit compositions and regulatory properties along with individual tissue and membrane distribution [40]. While low levels of several NOX enzymes are expressed in many tissues and cell types, NOX1 is predominantly expressed in the colon and prostate, NOX2 in myeloid cells, including granulocytes, monocytes and macrophages, NOX3 in the inner ear, NOX4 in the kidney, NOX5 in the spleen and testis, and DUOX1-2 in thyroid tissue [38,40]. Detailed aspects of the tissue distribution, function, and cancer relevance of NOX enzymes are described elsewhere [3].

NOX1, NOX2 and NOX3 comprise membrane-bound as well as cytosolic subunits that are separated at resting conditions. Upon activation, the cytosolic subunits merge with the membrane-bound subunits, which leads to enzyme activation and superoxide production. NOX4 is composed of the two membrane-bound subunits NOX4 and p22^phox^ and is constitutively active in contrast to other NOX. However, its activity is further enhanced by factors such as protein disulfide isomerase, or the Polymerase-δ Interacting Protein 2 Factor, which are co-localized with the NOX4/p22^phox^ complex [40]. The activity of the NOXs DOUX1 and DOUX2 is regulated by calcium [8,41].

## 5. NOX in Cancer and Their Regulation of PI3K Signaling

The primary function of NOX enzymes is to produce ROS with an important role in redox regulation by oxidizing thiol groups on proteins and thereby modulating their function and activation status. In addition, NOX-derived ROS participate in cell-type-specific physiological functions, such as microbial killing by phagocytes and production of thyroid hormones by follicular cells. However, if inadequately regulated, NOX-derived ROS may cause pathological oxidative stress and damage to DNA, lipids and other cell components [3].

Redox regulation in cancer is critical, since the activity of several cancer related enzymes including RAS, AKT and PTEN and tumor suppressor gene products including p53 and Rb are regulated by thiol activation [7,40]. As shown in Figure 1, thiol oxidation of PTEN and protein tyrosine phosphatases inactivates these enzymes, which enhances signaling via the PI3K-pathway. Furthermore, elevated ROS levels promote the nuclear translocation of NF-κB and HIF-1α, which activate several genes involved in growth and survival of cancer cells [7,42]. Though an aberrant oxidative environment has been linked to evolvement of both solid tumors and hematological malignancies, excessive ROS may also cause apoptosis of malignant cells and thereby limit cancer progression. Additionally, the use of antioxidants in cancer has generated variable results suggesting that the role of ROS in cancer may be context dependent [3,8].

NOX1 is highly expressed in the colon and NOX1-derived ROS has been implicated in development of colon cancer by mechanisms involving enhanced tumor cell proliferation and metastasis [43]. Ohata et al. demonstrated that NOX1-induced ROS promoted proliferation of cancer stem cell-enriched colon spheroids from clinical specimens via activation of mTORC1 kinase, highlighting the positive feedback of NOX-derived ROS and the PI3K pathway [44].

NOX2 is expressed in mature myeloid leukemia cells [45] and extracellularly released NOX2-derived ROS were shown to induce apoptosis of adjacent anti-leukemic lymphocytes to thus prevent destruction of malignant cells [11,46,47]. Furthermore, leukemic stem cells express low levels of NOX2, which appears critical for self-renewal. Hence, NOX2 inhibition in myeloid leukemia may reduce leukemic burden [45,48,49]. NOX2 was also found to be overexpressed in other histotypes of human cancer including diffuse large B cell lymphoma, breast invasive carcinoma, lung adenocarcinoma and non-small-cell lung cancer where it may either promote tumor growth or mediate tumor cell apoptosis [43,50,51].

NOX4-induced ROS regulate a variety of signaling pathways including PI3K/Akt, NF-κB, and STAT3, which may promote growth, metastasis and chemoresistance of tumor cells [52,53]. NOX4 is overexpressed in several types of cancer including breast cancer and kidney cancer, non-small cell lung cancer (NSCLC) where ROS generated by NOX4 promote cell proliferation and cancer metastasis [52,54,55]. In an experimental model of NSCLC, overexpression of NOX4 yielded larger tumors and pronounced lung metastasis. Inhibition of the PI3K pathway by LY294002 or wortmannin blocked the cellular effects of NOX4 overexpression in NSCLC cells both in vitro and in vivo, implying a positive feedback loop between NOX4 and PI3K/Akt signaling contributes to NSCLC progression [52].

Another study in NSCLC showed that NOX4-derived ROS from NSCLC cells stimulate PI3K/AKT signaling-dependent production of cytokines and chemokines leading to recruitment of tumor-promoting M2 macrophages. These events were inhibited by the PI3K inhibitor LY294002 or the Akt inhibitor MK2206 [56]. An additional NSCLC study demonstrated positive feedback between NOX4-derived ROS and AKT signaling on IL-6 production, which contributed to proliferation and survival of NSCLC cells [57]. NOX5 is overexpressed in clinical esophageal squamous cell carcinoma tumors, where NOX5-induced ROS may promote tumor development [58]. NOX3 has thus far not been linked to malignancy [43].

## 6. Regulation of NOX Enzymes by the PI3K Pathway

The role of PI3K/AKT signaling in activation of NOX enzymes has been suggested by inhibition of NOX activity and ROS production by pharmacological PI3K inhibition using wortmannin, a specific inhibitor of PI3K or by genetic inhibition by knockout of AKT1 [59,60]. NOX2 is highly expressed by myeloid cells including monocytes and neutrophils and is also expressed in vascular endothelial cells [61]. NOX2 comprises the membrane-bound subunits gp91^phox^ (aka NOX2 or CYBB) and p22^phox^ (aka CYBA) along with the four regulatory subunits p40^phox^ (aka NCF4), p47^phox^ (aka NCF1), p67^phox^ (aka NCF2) and the GTPase Rac, which are localized in the cytoplasm under resting conditions. Distinct subcellular localization of these two groups of components ensures that the oxidase remains inactive in resting cells [3,62]. When a resting cell is exposed to any of a variety of stimuli including pathogen-associated molecular patterns, bacterial peptides, growth factors, cytokines and immobilized IgG antibodies, the cytosolic component p47^phox^ becomes phosphorylated and the entire cytosolic complex translocates to the membrane where it assembles with gp91^phox^ and p22^phox^, thus forming a functional NOX2 complex that leads to the production of superoxide.

Several studies show that the phosphorylation of the cytosolic subunit p47^phox^ is indispensible for NOX2 activation [62,63]. For example, monocytes from patients with p47^phox^-deficient chronic granulomatous disease do not generate ROS via NOX2 and are therefore highly susceptible to bacterial and fungal infections as well as granuloma formation and sterile hyperinflammation in tissues [39,47]. Different pathways, including protein kinase C (PKC) stimulation and PI3K activation have been shown to trigger p47^phox^ phosphorylation and NOX2-derived ROS production [5,6,7,64]. Hence, AKT, the main downstream target of PI3K, phosphorylates the p47^phox^ at Ser304 and Ser328 and thus contributes to the initiation of ROS production [15,65]. While phosphorylation of p47^phox^ has been most extensively studied, several other NOX2 subunits have been shown to become phosphorylated following stimulation with PMA and bacterial products such as formyl-Met-Leu-Phe (fMLF), which swiftly trigger NOX2 activity [66]. Of interest, PIP3, the product of PI3K activation, was shown to directly interact with PX-domain of p40phox [67,68]. Also, PI3K-dependent activation of Rac2 has been reported [69]. Hence, there are several potential mechanisms that link PI3K stimulation and NOX activation in addition to AKT-dependent p47^phox^ phosphorylation.

NK cells and myeloid cells express Fc gamma receptors (FCγR) that attach the Fc portion of IgG to exert antibody-dependent cellular cytotoxicity (ADCC) or phagocytosis of foreign cells. In addition to enhancing ADCC, IgG antibodies, including therapeutic anti-CD20 antibodies used in the treatment of CLL, trigger neutrophils and monocytes to release extracellular NOX2-derived ROS via FCγR activation. This antibody-induced burst has been shown to compromise the survival and function of adjacent NK cells [70,71]. Our group recently assessed the role of PI3K isoforms, in particular PI3Kδ, in Fc-antibody-induced ROS formation. The results showed that the PI3Kδ inhibitor idelalisib prevented anti-CD20 induced NOX2-derived ROS production in human monocytes, as well as antibody-induced Akt phosphorylation and phosphorylation of the p47^phox^ subunit of NOX2 [15] (Figure 2).

Additionally, in monocyte/NK cell cocultures, idelalisib efficiently prevented ROS-inflicted NK-cell cytotoxicity, which entailed markedly improved NK-cell mediated ADCC against primary and cultured CLL cells [15]. Similarly, idelalisib inhibited ROS production from murine myeloid cells and improved NK cell-dependent clearance of metastatic cells in wild type mice, but not in *Nox2* knockout mice [15]. These results imply that idelalisib exerts antineoplastic effects in vivo by targeting NOX2 beyond interference with BCR signaling in malignant B cells. PI3K p110δ inhibitors may also be effective in other forms of leukemia including acute myeloid leukemia [11] and in chronic myelomonocytic leukemia [46], where NOX2-dependent immunosuppression may occur. In addition, p110δ inactivation in mice exerts anti-neoplastic activity in other cancers, including non-haematological solid tumors [15,72].

Other isoforms of class I PI3Ks have been reported to regulate NOX2-induced ROS production. For example, PI3Kβ inhibition using the selective inhibitor TGX-221 reduced NOX2-derived ROS production by integrin-activated neutrophils [73] implying that PI3Kβ plays an important role in NOX activation in adherent neutrophils. In addition, the PI3Kγ isoform is involved in NOX2 activation and ROS production by neutrophils stimulated by the GPCR ligand fMLF. Hence, fMLF-induced ROS-production was prevented by a PI3Kγ selective inhibitor [5].

Increased NOX activity and ROS production play a central role in the pathogenesis of vascular disease [74]. NOX1 is composed of the membrane-bound subunits NOX1 and p22^phox^ and the cytosolic subunits p47^phox^ homolog NOX organizing protein 1 (Noxo1), the p67^phox^ homolog NOX Activating protein 1 and Rac that assemble with the plasma membrane subunits upon activation [75]. It was reported that Noxo1 lacks the phosphorylated region found in its homolog p47phox and therefore the current view is that NOX1 activation occurs without Noxo1 phosphorylation. However, a recent report shows that Noxo1 phosphorylation on serine 154 is critical for optimal NOX1 assembly and activation [76]. NOX1 is highly expressed in vascular smooth muscle cells and has been implicated in vascular contraction and blood pressure elevation by peptide hormone angiotensin II. This peptide activates PI3Kγ via G protein βγ subunit (Gβγ), leading to the initial activation of NOX1, which is responsible for the first phase of ROS production and the ensuing calcium ion (Ca^2+^) influx, being involved in contraction [77]. The PKCβ signaling axis then mediates the second phase of ROS generation by NOX1. Inhibition of PI3K by wortmannin or by the PI3Kγ-specific inhibitor AS252424 inhibited the initial ROS production by NOX1 and attenuated the Ca^2+^ elevation in vascular smooth muscle cells after stimulation with angiotensin II [71]. This implies that PI3Kγ signaling may be a potential target for antihypertensive therapeutic intervention. NOX1 is also expressed in vascular progenitor cells. Furthermore, in vitro studies have shown that the PI3Kα isoform is involved in regulation of NOX1-induced ROS production and vasculogenesis in mouse embryonic stem cells stimulated with vascular endothelial growth factor, implying that the PI3Kα signaling pathway represents a promising therapeutic target for the treatment of diseases that involve aberrant neovascularization [78]. NOX1 is also expressed in colon tissues and has been implicated in pathophysiology of colon cancer and blocking of the epidermal growth factor receptor/PI3K/Akt signaling pathway reduced NOX1-induced ROS [44,79].

## 7. Conclusions

In cancer, many pathways may entail over-activation of the PI3K/AKT signaling, and it is becoming increasingly evident that NOX-derived ROS production may contribute to cancer tumorigenesis at least in part via stimulating the PI3K/AKT pathway. Several lines of evidence imply that PI3K/AKT signaling may directly activate at least NOX1 and NOX2-derived ROS. For NOX2 activation, this is at least in part mediated by AKT-induced phosphorylation of the regulatory subunit p47^phox^, although additional pathways have also been suggested. Whether or not PI3K promotes and additionally activates other NOX isoforms needs to be further investigated. A better understanding of the relationship between PI3K isoforms and NOX-derived ROS is relevant for development of therapeutic strategies to target tumors dependent on this pathway.

## Figures and Tables

**Figure 1 antioxidants-12-00067-f001:**
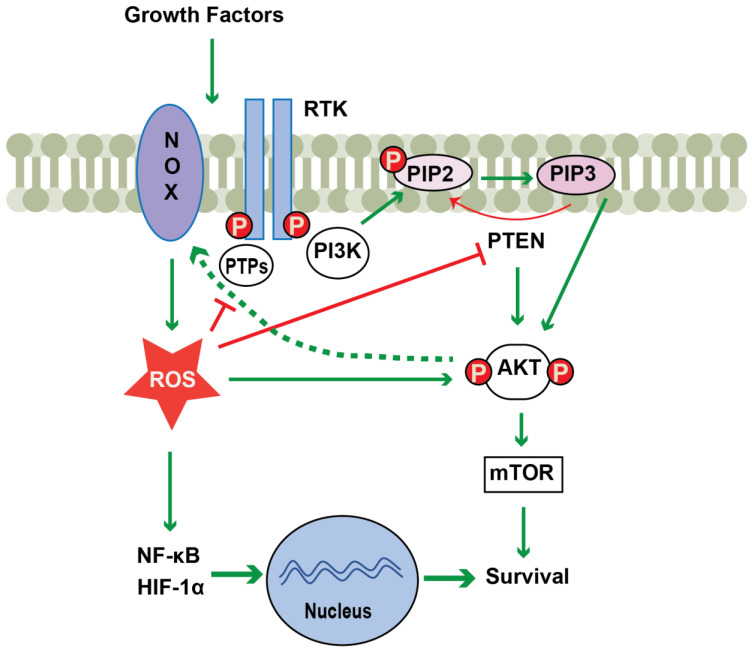
Schematic representation of NOX regulation of the PI3K/AKT pathway. Activation of NOX occurs upon, for example, growth factor stimulation. NOX-derived ROS potentiate the PI3K signaling cascade by directly activating AKT, or indirectly through inhibition of phosphatases such as PTEN or PTPs. In addition, NOX-derived ROS induce the nuclear translocation of NF-κB and HIF-1α, which promotes cell survival. ROS-mediated activation of AKT also serves to enhance NOX-derived ROS production in a positive feedback loop (dashed green arrow). Abbreviations: RTKs, receptor tyrosine kinases; PI3K, phosphoinositide 3-kinase; PIP2, phosphatidylinositol-4,5-bisphosphate; PIP3, phosphatidylinositol-3,4,5-trisphosphate; AKT, protein kinase B; PTP, protein tyrosine phosphatase; PTEN, phosphatase and tensin homolog; Green arrows and green dotted arrow indicate activation and red block arrows indicate inhibition; P, phosphorylation.

**Figure 2 antioxidants-12-00067-f002:**
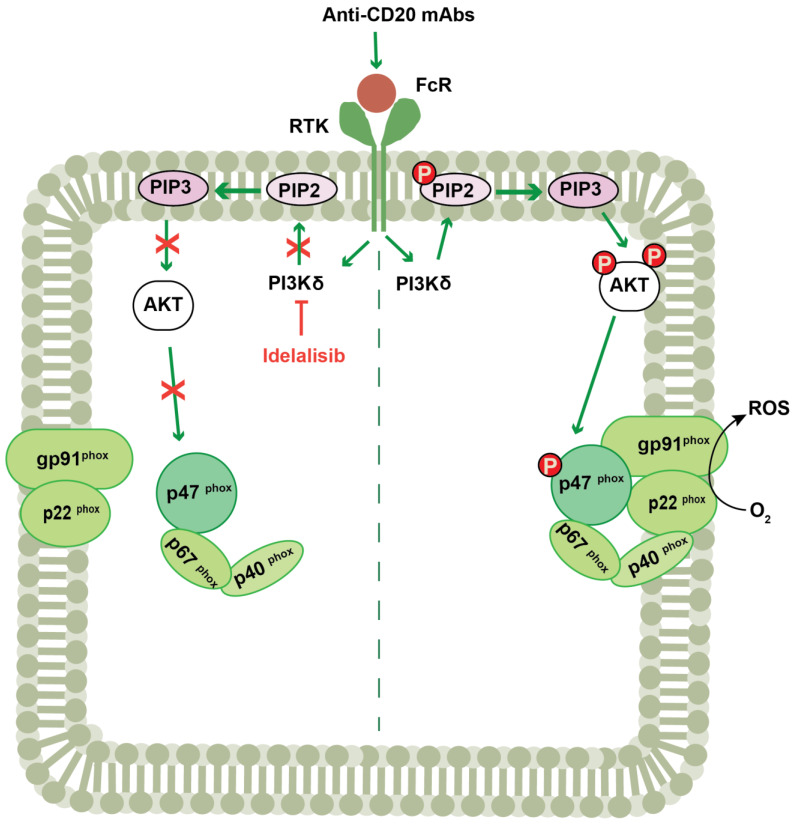
Schematic representation of the interplay between PI3K p110δ signaling and NOX2. Exposure of monocytes and neutrophils to a variety of stimuli, including immobilized IgG antibodies, activates NOX2 to generate ROS. IgG antibodies bind to Fc gamma receptors (FCγR) on the surface of monocytes and activate PI3Kδ, which in turn phosphorylates AKT. Upon activation, AKT phosphorylates p47^phox^ subunit of NADPH oxidase and then the entire cytosolic complex translocates to the membrane, where it assembles with gp91^phox^ and p22^phox^, thus forming the NADPH–oxidase complex, which leads to the production of ROS. PI3K p110δ inhibitor idelalisib inhibits the formation of NOX2-derived ROS. Abbreviations: RTK, receptor tyrosine kinase; PI3K, phosphoinositide 3-kinase; PIP2, phosphatidylinositol-4,5-bisphosphate; PIP3, phosphatidylinositol-3,4,5-trisphosphate and AKT, protein kinase B.; ROS, reactive oxygen species. Green arrows indicate activation. Red arrow and red “X” shaped symbol indicate inhibition. P, phosphorylation.

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
