# Peer review of "Role of Phosphoinositide 3-Kinase in Regulation of NOX-Derived Reactive Oxygen Species in Cancer"

_antioxidants, 2022, doi:10.3390/antiox12010067_

Round 1

Reviewer 1 Report

The manuscript is a review article describing the role of PI3K in regulation of NOX-derived ROS generation in cancer. This is one of the key non-enzymatic cellular mechanisms through which ROS generation can lead to the initiation and development of cancer. The manuscript provides and insight into the role of PI3K in NOX-derived ROS generation in cancer.  The review is well prepared, up-to-date and provides valuable insight. There are few suggestions for improvement.

1.       The authors should provide the importance of this pathway and its canonical role during infection and inflammation.

2.       The connection between inflammation and cancer development needs additional discussion integrating the role of PI3K/Akt.

3.       A table highlighting various NOX family members in various organs and cell types and their role in cancer development will be helpful to the readers.

Author Response

Reviewer 1

Comments and Suggestions for Authors

The manuscript is a review article describing the role of PI3K in regulation of NOX-derived ROS generation in cancer. This is one of the key non-enzymatic cellular mechanisms through which ROS generation can lead to the initiation and development of cancer. The manuscript provides and insight into the role of PI3K in NOX-derived ROS generation in cancer.  The review is well prepared, up-to-date and provides valuable insight. There are few suggestions for improvement.

  1. The authors should provide the importance of this pathway and its canonical role during infection and inflammation.

Response: We thank the reviewer for the positive feedback and helpful comments. As suggested, the revised manuscript includes a section denoted “Role of PI3K in regulating the tumor microenvironment and inflammation”, where the role of the PI3K pathway in infection and inflammation is briefly discussed (line 77).

  1. The connection between inflammation and cancer development needs additional discussion integrating the role of PI3K/Akt.

Response: A discussion regarding inflammation and cancer is included in the revised manuscript, line 102.

  1. A table highlighting various NOX family members in various organs and cell types and their role in cancer development will be helpful to the readers.

Response: While this is a good idea, our group included such a table in a recent review (Grauers Wiktorin, Oxid Med Cell Longev, 2020). In the revised manuscript we are referring to this review for detailed information of tissue distribution, function and cancer relevance of the various NOX enzymes (line 129).  

Reviewer 2 Report

Overall impression

The manuscript by Akhiani and Martner aims to review the most recent literature on the role of the PI3-K/AKT pathway on the activation of NOX complexes and subsequent induction of ROS in the context of lymphoid and epithelial carcinogenesis. Overall, the manuscript is well written, it is nice to see that it has focused upon a specific part of the literature, and it also makes use of appropriate schematics.

However, I believe that Figure 1 can be improved. Also, there is a small number of inaccuracies in some places, and a few minor issues with the figures (relating to usage of nomenclature, which needs to be more conventional).   

Minor points

1. The schematic in Figure 1 could be improved:

i) it is currently unidirectional with regards to NOX-PI3K/AKT interactions as it visualises the consequence of NOX induction on the PI3K/AKT axis but not vice versa. I appreciate that to some extent this is visualised in Figure 2; however, it is a shame that Figure 1 does not take into account the bidirectional nature of the interactions, i.e. role of PI3K/AKT on NOX (in line with the article’s title). Either modify Figure 1 itself, or at least make a comment in the main text that this is indicated in Figure 2?  

ii) it is also worth indicating in the caption the usage of red and green colours for the arrows (inhibition vs activation)

2. In Figures 1 and 2, please use more conventional terminology and/or abbreviations:

i) in both figures, please use the term RTK (receptor tyrosine kinase) and not TKR,

ii) in Figure 1, correct “NF-kappa-beta” (currently with two symbols) to “NF-kappaB” (capital B not beta), and also amend “HIF1-alpha” to the more conventional term “HIF-1alpha”

3. In the Reference list, I do not believe the last reference (#73) has been cited in the text. Also, please check ref#12 for possible issue with accuracy / formatting?  

4. Please check the text for a few minor typos:

a) Line 113: “NOXes” should be “NOXs” 

b) Line 130: please correct the terms “NF-kappa-beta” and “HIF1-alpha” (as per my comment above on the figures)  

c) Line 149: please correct “histiotypes” to “histotypes”?

d) Line 214: please check the word “comprome”… is it supposed to read “compromise”…?

Author Response

Reviewer 2

Comments and Suggestions for Authors

Overall impression

The manuscript by Akhiani and Martner aims to review the most recent literature on the role of the PI3-K/AKT pathway on the activation of NOX complexes and subsequent induction of ROS in the context of lymphoid and epithelial carcinogenesis. Overall, the manuscript is well written, it is nice to see that it has focused upon a specific part of the literature, and it also makes use of appropriate schematics.

 However, I believe that Figure 1 can be improved. Also, there is a small number of inaccuracies in some places, and a few minor issues with the figures (relating to usage of nomenclature, which needs to be more conventional).  

Minor points

  1. The schematic in Figure 1 could be improved:
  2. i) it is currently unidirectional with regards to NOX-PI3K/AKT interactions as it visualises the consequence of NOX induction on the PI3K/AKT axis but not vice versa. I appreciate that to some extent this is visualised in Figure 2; however, it is a shame that Figure 1 does not take into account the bidirectional nature of the interactions, i.e. role of PI3K/AKT on NOX (in line with the article’s title). Either modify Figure 1 itself, or at least make a comment in the main text that this is indicated in Figure 2?  

Response: We thank the reviewer for the positive feedback and helpful comments that have served to improve our manuscript. We appreciate this comment and have updated the Figure 1 and its figure legend (line 310) to include also the positive regulation of phosphorylated AKT on NOX-derived ROS production.

  1. ii) it is also worth indicating in the caption the usage of red and green colours for the arrows (inhibition vs activation)

Response: Edited as suggested (Figure 1 and line 314).

  1. In Figures 1 and 2, please use more conventional terminology and/or abbreviations:
  2. i) in both figures, please use the term RTK (receptor tyrosine kinase) and not TKR,
  3. ii) in Figure 1, correct “NF-kappa-beta” (currently with two symbols) to “NF-kappaB” (capital B not beta), and also amend “HIF1-alpha” to the more conventional term “HIF-1alpha”

Response: Changed as suggested.

  1. In the Reference list, I do not believe the last reference (#73) has been cited in the text. Also, please check ref#12 for possible issue with accuracy / formatting?  

Response: Reference #73 has been cited in the Figure 1 legend (line 310). Reference 12 is formatted in the revised manuscript.

  1. Please check the text for a few minor typos:
  2. a) Line 113: “NOXes” should be “NOXs” 

Response: Changed as suggested (line 139).

  1. b) Line 130: please correct the terms “NF-kappa-beta” and “HIF1-alpha” (as per my comment above on the figures)  

Response: Changed as suggested (line 155).

  1. c) Line 149: please correct “histiotypes” to “histotypes”?

Response: Changed as suggested (line 174).

  1. d) Line 214: please check the word “comprome”… is it supposed to read “compromise”…?

Response: Changed as suggested (line 238).

Reviewer 3 Report

The authors have written an interesting review on the role of Phosphoinositide 3-kinase (PI3K) in the regulation of NOX-induced ROS formation in cancer. However, I have the following comments to improve the review:

 Major:

 1.    Authors are suggested to add a separate section on the role of PI3K in regulating tumor microenvironment.

2.    Authors are suggested to add a detailed separate section on the therapeutic strategies to regulate PI3K signaling to regulate cancer progression preferably using a table.

 Minor:

 1.    Authors are suggested to correct the typological mistakes. e.g., in line 12, “phoinositide 3-kinase” should be “Phosphoinositide 3-kinase”.

2.    Authors are suggested to incorporate keywords.

Author Response

Reviewer 3

Comments and Suggestions for Authors

The authors have written an interesting review on the role of Phosphoinositide 3-kinase (PI3K) in the regulation of NOX-induced ROS formation in cancer. However, I have the following comments to improve the review:

Major:

  1. Authors are suggested to add a separate section on the role of PI3K in regulating tumor microenvironment.

Response: We thank the reviewer for the positive feedback and helpful comments. As suggested a new section denoted “Role of PI3K in regulating the tumor microenvironment and inflammation” has been included in the revised manuscript (line 77).

  1. Authors are suggested to add a detailed separate section on the therapeutic strategies to regulate PI3K signaling to regulate cancer progression preferably using a table.

Response: The revised manuscript briefly discusses therapeutic strategies to regulate PI3K signaling in cancer (line 89 and line 105). In addition, we refer readers to a comprensive recent review of therapeutic strategies to modulate PI3K in cancer (He et al, Signal Transduct Target Ther 2021) (line 108).

Minor:

  1. Authors are suggested to correct the typological mistakes. e.g., in line 12, “phoinositide 3-kinase” should be “Phosphoinositide 3-kinase”.

Response: Phosphoinositide 3-kinase is correctly written (line 11).

  1. Authors are suggested to incorporate keywords.

Response: The revised manuscript includes keywords (line 15).

Round 2

Reviewer 1 Report

The authors have satisfactorily revised the manuscript and addressed the comments. No further concerns.